# 'A Most Select Gathering'. Mexican National Pilgrimages to Rome during the Papacy of Leo XIII

**Francisco Javier Ramón Solans** [1,2]

1 Department of History, University of Zaragoza, 50009 Zaragoza, Spain; fjramon@unizar.es
2 Institute of Heritage & Humanities, University of Zaragoza, 50009 Zaragoza, Spain

**Abstract:** The objective of this article is to analyse Mexican national pilgrimages to Rome that took place during the pontificate of Leo XIII (1878–1903). These pilgrimages occurred in the context of a global Catholic mobilisation in support of the papacy, during the so-called Roman Question. This paper's analysis of these pilgrimages draws from historiography about national pilgrimages, as well as studies on Catholic mobilisation in support of the pope in the second half of the nineteenth century. It is fundamentally based on primary sources of an official nature, such as reports and other printed documents produced on the occasion of the pilgrimage. The study's primary conclusion is that national pilgrimages to Rome had a polysemic character since they brought together various religious and national identities. The pilgrimages contributed simultaneously to reinforcing the link between Catholicism and Mexican national identity and the global dimension of Catholicism and allegiance to the Holy See.

**Keywords:** catholic pilgrimages; Mexican Catholicism; papacy; Roman Question; ultramontanism; Latin America





## 1. Introduction

In the nineteenth century, the practice of pilgrimage was profoundly and lastingly reshaped, so much that in many respects it still pertains in shrines around the world. This model was characterised by its large-scale nature, enabled by the incorporation of means of transport such as railways or steam ships, communication media such as newspapers, and mass-produced commodities such as small medals, prayer cards, and other types of souvenir (Cinquin 1980; Harris 1999; Kaufman 2005; Wharton 2006; Eade 2015). The need for an organisational structure capable of planning the journey and accommodating such a large number of pilgrims daily, as well as the new consumer and recreational relationships the pilgrims established with their destination, were present in the rise of the new figure of the pilgrim-tourist and of mass tourism itself (Dupront 1967; Kark 2001; Cohen-Hattab and Shoval 2003; Cohen-Hattab and Shoval 2015; Timothy and Olsen 2006; Raj and Morpeth 2007). These new devotional models spread rapidly across the globe (Kotulla 2006; Halemba 2008; Ramón Solans 2016) and later had an important impact on the development of secular pilgrimages, as well as in the reinvention of sacred places (Reader and Walter 1993; Margry 2008; Barbato 2013; Eade 2020).

This regeneration of devotional cultures involved not only the increase in individual pilgrimages but also the rise in new forms that were characterised by their collective nature, whether that was national or regional, and which played a central role in Catholic mobilisation from the middle of the nineteenth century until the Second Vatican Council. In the framework of the so-called 'culture wars' that questioned the role of religion in the public sphere (Clark and Kaiser 2003; Weir 2018), pilgrimages earned a new political meaning, as the incarnation of the true national community, as well as a means of protesting against the implementation of secularising measures.

Although we already have some overviews of these mobilisations (Pazos 2020; Di Stefano and Ramón Solans 2016; Chantre et al. 2014; Zimdars-Swartz 1991), analysis of

these pilgrimages has largely been subsumed into the more general study of a devotion (Boutry and Cinquin 1980; Blackbourn 1993; Christian 1996; Cenarro Lagunas 1997; Serrano 1999; Harris 1999; Jonas 2000; de la Cueva 2001; Di Febo 2002; Hall 2004; Boyd 2007; Hynes 2009; Ramón Solans 2014) or has focussed on prominent pilgrimages such as those of the Trier Holy Tunic which attracted half a million people in 1844 (Schieder 1974; Lill 1978; Schneider 1995) or on periods of intense mobilisation such as the French pilgrimages launched in 1873 (Maës 2011). These works emphasise the mobilising aspect of Catholic devotions, their capacity for driving protests against secularist measures or governments, and their role in the construction of national and regional identities.

The disappearance of the Papal States with the capture of Rome in 1870 stirred a wave of protests in the Catholic world, which was channelled, among other elements, through the celebration of national pilgrimages to Rome in solidarity with the pope. From the mid-nineteenth century, the papacy's progressive loss of territories in the face of advancing Italian unification awoke a wave of empathy and solidarity with the papacy, which came to be expressed in forms of devotion to the pope as an alter christus (Horaist 1995; Zambarbieri 2005; Seiler 2007). The national pilgrimages had a strong political dimension, both in their defence of the papacy's temporal sovereignty as well as for their nature of protest against the participants' political regimes, such as the Royalist Catholics against the French Third Republic, the Legitimists against the Spanish Bourbon Restoration, or the German Catholics against the *Kulturkampf* (Brennan 2000; Dupont 2018; Heid 2020). As well as national pilgrimages, workers' pilgrimages were also organised which, financed by parishes and businesses, served to demonstrate the adhesion by the proletariat to Catholicism and thereby to fight socialism and anarchism. Researchers have particularly focussed on the French and Spanish pilgrimages (Brennan 2000; Faes Díaz 2009). In turn, the Holy See contributed to these types of national and worker pilgrimages with the celebration of ordinary and extraordinary jubilees which illustrated the global dimension of papal power (Ticchi 2005). Despite the works mentioned, however, there are no overall studies of the national pilgrimages that arrived in Rome during this period, nor have the Latin American pilgrimages that took place during the pontificates of Pius IX and Leo XIII even been analysed.

Throughout the second half of the twentieth century, mass pilgrimages diminished since many of those devotions depoliticised with the development of tourism in folklorised versions, diluted into forms of religiosity without culture, more fluid and individual (de Certeau and Domenach 1974; Roy 2008; Pack 2010). Social sciences have undergone a similar evolution when it comes to addressing the meaning of the pilgrimages. Early studies cemented the idea that they served to unite and reinforce identities and social hierarchies (Durkheim [1909] 2007; Wolf 1958). This seems logical given that in Émile Durkheim's time, pilgrimages were characterised as collective, rigid, and hierarchical, in the context of a struggle for public space and a fight against secularisation (Fournier 2007). In contrast to this model, and in a context of depoliticization of religious practices, other authors proposed the idea that pilgrimage was a liminal phenomenon, that far from reinforcing a social and ideological structure, it temporarily suppressed it, giving rise to an egalitarian and level *communitas* among those present (Turner and Turner [1978] 2011). To overcome this dichotomy, it has been suggested that pilgrimages may be understood as 'an arena for competing religious and secular discourses, for both the official co-optation and the non-official recovery of religious meanings' (Eade and Sallnow 1991, p. 2).

To address the multiple meanings that are observable in these devotional practices, the objective of this article is to study the three Mexican national pilgrimages to Rome during the pontificate of Leo XIII (1878–1903). Despite the prominence that they had at the time and the importance accorded to them by their participants, however, these pilgrimages have not been the subject of an historiographical study, either individually or collectively, but have instead been subsumed into the study of Porfirismo (Valadés 2015, pp. 603–6) or of Mexicans' travels in the nineteenth and twentieth centuries (Teixidor 1939). The working hypothesis is that these pilgrimages were characterised by being polysemic and

ambiguous since they combined various superimposed Catholic identities, almost like a Matryoshka doll: global, Latin American, national, and regional. In fact, a majority of the actors who organised the pilgrimage had a clear transnational profile, since despite being Mexican, they had trained at the Pontifical Latin American College in Rome and already had experience of global or supranational events in Europe. Finally, although it was through a dynamic of central and peripheral, these changes contributed to increasing the representation of the Latin American space in global Catholicism.

## 2. Results

### 2.1. Historical Context

To explore the multiple global identities that mixed during the pilgrimage, we will focus on the Mexican national pilgrimages organised in 1888, 1897, and 1900. The investigation is based on the reports that were written during and after the pilgrimage to tell the other side of the Atlantic about them, convey feelings and experiences, specify the multiple meanings of the events, and serve as a memento of the pilgrimage itself. From a methodological perspective, this investigation is based on the methods of cultural history, religious history, and global history. Before analysing the three pilgrimages, and in order to better understand the multiple identities at play, it is necessary to situate them within the triple religious context of Mexico, Latin America, and global Catholicism.

### 2.1.1. The Religious Reform of Mexico

From the middle of the nineteenth century, the Mexican church experienced a period of violent secularisation. After the overthrow of the dictator Santa Anna in 1855, Mexico remained mired in a state of political instability that culminated in the Reform War (1858–1861) between liberals and conservatives. In this context, the presidencies of Ignacio Comonfort (1855–1858) and Benito Juárez (1858–1872) drove important secularising measures known as the Reform Laws: the expulsion of the Jesuits (1856), Lerdo's confiscation (1856), the secularisation of marriage, the civil registry, and cemeteries (1859), or the freedom of worship (1860). These measures prompted several protests in the Catholic media, which resulted in the exile and the confinement to the capital of several Mexican bishops (Brading 2000). Paradoxically, this very hostile media favoured reform in the Mexican church by making impossible any Gallicanist solution that supported a 'national' church protected by the state. This encouraged the adoption of Ultramontane postulates that led to the Romanisation of the Mexican church. In fact, many of those exiled prelates fled to Rome and, from there, advanced the training of Mexican priests in the Pontifical Latin American College (Brading 2000; Bautista García 2005; Edwards 2011; Bautista García 2012; Connaughton 2010; Mijangos y González 2015).

The lingering embers of the Reform War (1858–1861) were still smouldering when President Sebastián Lerdo de Tejada (1872–1876) incorporated the Reform laws into the Magna Carta and succeeded in elevating the separation between church and state to a constitutional level. These measures produced discontent in the Catholic media and even fed a military uprising known as the Religionero Rebellion (1873–1876). The nature of the Religionero movement cannot simply be reduced to a question of church-state, however, since it did not include clergy, and the Mexican episcopate condemned all forms of violence against the government in 1875. Brian A. Stauffer (2020) has therefore pointed out that other factors should be taken into account, such as the popular discontent produced by changes in devotional cultures driven by Ultramontane sectors, especially by their attempt to control indigenous religious practices and the imposition of normative religious practices of European origin.

The general Porfirio Díaz—who had lost the previous elections against Juárez and Lerdo de Tejada—staged a coup d'état which gave him the presidency of Mexico, an office which he held almost without interruption from 1876 to 1911. Although he did not abolish any of the secularising measures during his tenure, he tempered their application or put them on hold. Porfirio Díaz also nurtured a good relationship with the ecclesiastical elite,

especially with the Archbishop of Oaxaca, Eulogio Gillow. In fact, this diocese represents a magnificent example of the progress of the Mexican church during the years of the Porfiriato: rising to the rank of archbishopric, hosting a Synod council in 1892, reorganising and expanding a better-trained local clergy, encouraging lay associations, etc. (Wright-Rios 2009).

During this period, the Mexican ecclesiastical hierarchy promoted a renewal of devotional cultures inspired by European models and especially by the Sanctuary of Our Lady of Lourdes. The clergyman José Antonio Plancarte y Labastida (1840–1898)—trained in Oscott and Rome and a pilgrim to Lourdes in 1877—founded the Congregación de Religiosas Hijas de María Inmaculada de Guadalupe in 1878 and fostered the coronation of the images of Nuestra Señora de la Esperanza in Jacona in 1885 and of Guadalupe in 1895 (Brading 2002). This devotional practice, highly developed in Italy and France, started to spread around the Catholic world and frequently entailed refurbishment of the church and a large regional or national pilgrimage (Langlois 2016).

As we have seen, the Mexican national pilgrimages to Rome which took place in the 1880s and 1890s occurred at an opportune time for this kind of huge demonstration of religious faith, since they had the approval if not the support of the civil authorities, as well as a renewed ecclesiastical and secular structure. These practices were likewise encouraged by a more disciplined and Romanised clergy, which had internalised the new devotional cultures promoted in Europe and had even gone to the Old World to train.

### 2.1.2. The Turn to Ultramontanism in Latin America

The 1840s and 1850s were marked by a huge advance in Ultramontane tenets at the heart of the different Latin American churches. Although this process of Romanisation of the Catholic Church was on a global scale, there are some specific features that allow us to explain its Latin American evolution, such as the tensions produced by excessive state intervention in religious matters, the impact of secular policies in Colombia and Mexico, and the promotion of an episcopal generation that was closer to Rome. These prelates played a fundamental role in the reform of their dioceses, promoting the foundation of new seminaries, the return of the Jesuits, the arrival of religious communities, the creation of spaces for lay participation (such as the Society of St Vincent de Paul), and the construction of a dense confessional publishing structure (Klaiber 1992; Brading 2000; Plata Quezada 2005; Serrano 2008; Martínez 2014; Santirocchi 2015).

This process of Romanisation was encouraged by the growing internationalisation of the Latin American ecclesiastical actors, who began increasingly to travel to the USA and Europe in search of new models to import into their countries. During the 1860s, the journeys of Latin American Catholics to Rome multiplied in the contexts of trips *ad limina*, exiles forced out by the political situation of their countries, and attendance at the great global Catholic events, such as the centenary of the martyrdom of St Peter and St Paul in 1867 and the First Vatican Council in 1870. These actors contributed to the spread of doctrines, congregations, new organisations, educational systems, publications, and campaigns in support of the pope in Latin America (Ramón Solans 2020a). They obviously contributed to the spread of new devotional cultures on the continent as well as to the renewal of centres of pilgrimage following, *mutatis mutandis*, the model of Lourdes sanctioned by Rome (Di Stefano and Mauro 2016; Hall 2016; Wright-Rios 2016; Monreal 2016).

The increased closeness of the various Latin American churches to Rome also contributed to increasing closeness to each other. The journeys the Latin American clergy made to the other side of the Atlantic often led them to cross other American countries and to meet clergy of other continental nationalities. Their growing internationalisation likewise meant that they encountered one another in other European spaces, especially in Rome. In their dialogues and epistolary exchanges, the Latin American ecclesiastical hierarchies soon understood that the challenges that they faced were similar and that it was useful to unite their efforts and strategies to overcome them. In the 1850s, the Archbishop

of Santiago, Chile, had already contacted some of his Latin American colleagues to try to organise a general council for the Central and South American countries. Although a council in Rome was not held until 1899, Valdivieso's correspondence shows the formation of a consciousness of collegiality and belonging to the same Latin American catholic space. This consciousness would be reinforced with the Plenary Council of Latin America (Ramón Solans 2020b, 2020c).

In turn, the Holy See thought out and enacted its strategies on the continent from a Latin American perspective (Cárdenas Ayala 2018). Especially important within this Latin American awakening was the creation in 1858 of the Pontifical Latin American College. Fostered by Pius IX and made reality through the efforts of the Chilean priest José Ignacio Víctor Eyzaguirre, who travelled the continent to find students and financing, this college sought to prepare the most brilliant youth of the Latin American dioceses in Rome so that upon their return they could modernise and Romanise their churches. Whether it was as bishops or teachers of diocesan seminaries, one of their main preoccupations was improving the education of the local priesthood, transmitting what they had learned in Rome about doctrine, discipline, spiritual practices, associations, devotions, and loyalty to the pope (Edwards 2011).

From the very beginning, and despite the Mexican episcopate finding itself in a very difficult situation as a consequence of the Reform War, it collaborated with the project. Among its early support, the Bishop of Puebla, Pelagio Antonio Labastida, stands out, promoting the college in his diocese from exile in Rome. He managed to achieve funding for two students, José Antonio Plancarte y Labastida, nephew of Archbishop Labastida and his right hand in the reform of the Mexican church, and Ignacio Montes de Oca, future Bishop of Tamaulipas (Brading 2002; Edwards 2011; Ramón Solans 2020c). Thanks to their Roman training and transnational experience, these priests trained in the Pontifical Latin American College played a very important role in the regeneration of devotional practices in Mexico and especially in the organisation of the first pilgrimages to Rome.

### 2.1.3. The Renewal of Pilgrimages to Rome

After the Napoleonic Wars, pilgrimages on foot to Rome almost disappeared. Between 1815 and 1865 the journey became the exclusive privilege of a small elite who could defray several months' stay abroad. It was the final swansong of the mythical Grand Tour that culminated in a visit to romantic Rome, that Rome immortalised by Goethe in his *Die italienische Reise* (1813–1817), destination of European Catholic nobles and writers who after the Revolution again turned their eyes to the capital of Christianity. It was a Rome restored not just politically but also artistically and archaeologically, with the expansion of the Vatican Library and Museums, the excavations in different parts of the Papal States, the restoration of ancient monuments such as the Colosseum, and the spectacular rediscovery of the Roman catacombs that reinforced the image of martyr Rome, the heart of Christianity (Boutry 1979; Viaene 2004).

The steady improvement of means of communication, with the foundation of rail networks in the north of Europe and steamship routes that connected Marseilles with the main Italian ports, made the journey rather more accessible, although it remained far beyond the means of the middle and lower classes. With the centralisation of Catholicism, Rome gradually became the destination for 'business trips' for priests, monks, nuns, and prelates as well as Catholic journalists, writers, and politicians who went to Rome in search of legitimisation, inspiration, and grants (Viaene 2001).

The revolution in Rome in 1848 and the Pope's exile in Gaeta prompted an unprecedented reaction in the Catholic world. In contrast to the bare echo produced by the imprisoning of Pius VI in 1799 and Pius VII in 1808 at the hands of French troops, the exile of Pius IX awoke a wave of solidarity across the globe. Among other factors, this reaction is explained by the improvement in communications, the crisis in the Gallican position, and the advancement of the Ultramontanist one. This outrage translated into innumerable letters, collections of money and signatures, and the issuing of a remarkable publicity in

support of the Pope. The first Latin American demonstrations of solidarity with the Pope arrived from Mexico, in particular with prime minister José Joaquín Herrera's invitation to Pius IX in 1849 to take refuge in Mexico, an invitation that was echoed by many of the country's city halls and prelates (Ramón Solans 2020a).

The loss of two thirds of Papal State territory as a consequence of the Italian war of 1859, and the rebellion of Romagna, Umbria, and the Marches drove the demonstrations of solidarity with the Pope. Five million signatures in support of Pius IX were amassed, 11,000 volunteers were recruited to defend the Pope, the collection of Peter's Pence was revived, and even devotion to the Pope was encouraged. These campaigns were encouraged by the Holy See and its diplomatic apparatus as well as by the Catholic press, episcopacy, lay associations, and so on (Horaist 1995; Guénel 1998; Viaene 2002; Pollard 2005). Despite the difficulties of the Mexican clergy during the Reform War, the episcopacy also demonstrated its loyalty to the Roman cause at that time and defended the temporal sovereignty of the Pontiff over his faithful (Poder Temporal 1861).

The Holy See promoted great ceremonies that enabled the expression of global support for the Pope. This was the case with the canonisation of the Martyrs of Japan in 1862 which drew 264 prelates from across the world, including the Mexican Bishops of San Luis de Potosí, Puebla, Michoacán, and Antequera, who found themselves exiled in Rome. The canonisation became a global protest against the unification of Italy and a demonstration of support for papal primacy. Something similar happened with the celebration of the XVIII centenary of the martyrdom of St Peter and St Paul in 1867, which drew 45 cardinals, 420 prelates, 18,000 clergy and 150,000 visitors to Rome from across the world (Viaene 2004; Riall 2010; Ramón Solans 2020a). Shortly before the fall of the Papal States, the first collective pilgrimage to Rome by German Catholics took place in 1869 (Heid 2020).

The conquest of Rome by Italian troops did not put an end to these demonstrations—in fact, quite the reverse. The pitiful image of a pope imprisoned and in chains was disseminated through various media in order to feed these demonstrations of affection, empathy, and solidarity with the pontiff even more. Festivals in 1871 and 1876 were organised for the 25th and 30th anniversaries of his elevation to the papacy, in 1877 for the 50th anniversary of his episcopal consecration, and in 1875 on the occasion of the jubilee. In the latter case, pilgrims were asked to arrive individually in Rome so as not to inflame sensitivities and cause a confrontation with Italian authorities. The pilgrimages organised during the final years of the papacy of Pius IX were characterised by the small number of participants, their elite composition, and their Legitimist stamp. This was the case for the first national French pilgrimage in 1873 and the Spanish one of 1876 (Brennan 2000; Kertzer 2004; Seiler 2007; Horaist 1995; Hibbs-Lissourgues 1995; Ramón Solans 2018; Heid 2020).

In the face of the Italian authorities' prohibition on making pilgrimages to the principal national sanctuaries in 1873, Catholics in Bologna inaugurated a new practice, spiritual pilgrimages, approved by Pius IX, which consisted of a mental journey that, through prayer and contemplation, would transport the pilgrim to the holy sites and most important national and international sanctuaries. In 1874 the Archbishop of Mexico, Pelagio Antonio Labastida y Dávalos, introduced this practice in his country to protest against the secularising measures rolled out by Lezo's government. This devotional practice also reinforced the Roman orientation of the Mexican church and encouraged the faithful to acquire an awareness of the global dimension of Catholicism and deepen their connection with the pope (Stauffer 2018).

During the papacy of Leo XIII, pilgrimages to Rome reached a new dimension in terms of both the numbers and the countries involved. As well as the ordinary jubilee of 1900, Leo XIII fostered various jubilees in 1879, 1882–1883, 1888, 1894, and 1902. The first great pilgrimage organised to Rome was the one celebrated in 1888 on the 50th sacerdotal anniversary of Leo XIII. The Bolognese publicist Giovanni Acquaderni played a central role in promoting these festivities, presiding over the organising committee of the jubilee and travelling throughout Europe to promote its celebration. The festivities were a success and between October 1887 and June 1888 some 44,194 pilgrims visited the city, the majority

(28,528) Italian (Ticchi 2005). The figures for the jubilee of 1900 are even more spectacular, with some 700,000 pilgrims from all over the world arriving during the festivities (Hilaire 2003).

The pilgrimages continued to be composed largely of the middle and upper classes, since both the journey and the accommodation was expensive and implied several weeks of travel. Within the context of the development of social Catholicism, however, some businessmen paternalistically financed the journeys of a large number of workers from the Catholic workers' circles. This was the case with the first pilgrimages of workers organised in 1885 and 1887 by the Archbishop of Reims Benoît Langénieux, as well as the Catholic entrepreneur Léon Harmel, and the French reformer Albert de Mun. The pilgrimage of 1887 was a great success and managed to bring 1400 workers, 110 factory owners, and 300 clergy to Rome. Two years later, in 1889, the pilgrimage reached 3000 workers and in 1891, three months after the publication of the *Rerum Novarum*, 20,000 French workers travelled to Rome in gratitude to Leo XIII for that document (Brennan 2000). In Spain, the Marquess of Comillas and Catholic industrialist, Claudio López Bru, promoted a workers' pilgrimage that brought 18,500 pilgrims to Rome in 1894 to celebrate the fiftieth anniversary of the priestly ordination of Leo XIII (Faes Díaz 2009).

Despite technological advances, the enormous distance that exists between Latin America and Europe practically constituted an insurmountable barrier to organising a massive and therefore expensive pilgrimage to Rome. While not as numerous as their European counterparts, during the papacy of Leo XIII, Latin American Catholics nevertheless began to organise their own mass pilgrimages to the Vatican to demonstrate their loyalty to the Holy See. Argentinian and Mexican Catholics took the lead. The continent's first mass pilgrimage was led by the provisor and vicar general, Antonio Espinosa, in Argentina in 1881, and later followed the Mexican ones of 1888 and 1898, and the pilgrimages of the jubilee of 1900 organised by Brazil, Argentina, Uruguay, Venezuela, and Mexico.

## 2.2. Mexican Pilgrimages to Rome during the Papacy of Leo XIII (1878–1903)

2.2.1. The First National Pilgrimage to Rome (1888)

In early 1887, news began to arrive in Mexico of the 'universal movement of the nations and of individuals to celebrate in a thousand ways the Sacerdotal Jubilee of Leo XIII which will take place on 1 January 1888' (German y Vázquez 1889, vol. 1, p. 3). The man promoting this initiative in Mexico was José María Mora y Daza, Bishop of Puebla de los Ángeles and an active priest who had reformed the *Seminario Palafoxiano* and driven a movement of monthly pilgrimages from the Mexican dioceses to the sanctuary of Our Lady of Guadalupe in February 1887. The idea was taken up by the Primate of Mexico, Pelagio Antonio Labastida, who issued a general call to Catholics across the whole country and ordered the formation of a propaganda committee in which lay people such as the pilgrimage reporter Diego Germán Vázquez would play a very important role. In order to let the Mexican people know about the pilgrimage, this committee wrote to the main towns in the country so they could publish an invitation to join the pilgrimage. To emphasise the Pope's authority, this text produced a play of contrasts that described 'an elderly man who, imprisoned in his house in one of the great cities of old Europe, is the object of the attention, the respect, the veneration of all peoples. A King without a state, he has, as subjects, millions of inhabitants around our world'. It likewise appealed to national pride for Mexicans to join 'this universal movement' by indicating that even the nations who 'are the greatest enemies of Catholicism, such as Turkey and China' were taking part in this jubilee (German y Vázquez 1889, vol. 1, pp. 6–7).

The unexpected death of José María Mora on 26 December 1887 was a huge blow for the pilgrimage and cancelling it was even considered. This bleak outlook was quickly banished when the prelate was taken up by the young vicar capitular of Puebla, Ramón Ibarra. This Mexican priest is a paradigmatic example of the importance of the internationalisation of the Latin American clergy for the Romanisation and reform of the various national churches. After training in the Seminario Palafoxiano, Ramón Ibarra attended the

Pontifical Latin American College in Rome and received a doctorate from the Pontifical Gregorian University in Philosophy, Theology, and Canon Law. Ramón Ibarra had already demonstrated his organisational abilities and concern for devotional renewal when he organised the first diocesan pilgrimage to Guadalupe in 1887. He not only managed to ensure the success of the first Mexican pilgrimage to Rome but also, then as Bishop of Puebla, played a central role in the two following pilgrimages to Rome in 1897 and 1900.

Before departing, the pilgrims congregated in the capital to attend a religious event in honour of the Virgin of Guadalupe. They thereby symbolically connected Mexican national devotion with Rome through 'the Mexican Virgin, Mexico's special protector, [who] would be the pilgrims' guide; beneath her shelter and protection they undertook the journey' (German y Vázquez 1889, vol. 1, p. 30). On 15 April 1888, the pilgrimage departed by rail from the capital to New York. At stations all along their route through Mexico, the pilgrims were celebrated and received with the national anthem.

From New York, the pilgrims embarked for Naples, whence they travelled by train to Rome. The pilgrimage was presided over by the Bishop of Chilapa, Buenaventura Portillo y Tejeda, and included representatives of the different Mexican dioceses, such as the canon of the Cathedral of Mexico City, Ambrosio Lara, and the canon and rector of the Seminary of Morelia, Agustín Abarca. In total, there were 150 pilgrims comprised of clergy, laymen, and three students who were being taken to the Pontifical Latin American College. Although we cannot construct an exact profile of the lay pilgrims, we can highlight that they belonged to the Mexican social elite, including lawyers, doctors, merchants, landowners, industrialists, engineers, representatives of lay organisations and Catholic press, etc. The pilgrimage included a small number of women—29 from Mexican high society—who largely travelled as wives, sisters, and daughters of the lay pilgrims. Although presented as a pilgrimage of all social classes, it lacked the presence of the masses as few could afford such a journey, and it was instead more of an occasion to identify oneself as elite and make a public demonstration of personal religious commitment.

Nor was the pilgrimage much more diverse in terms of race, since it included only one elderly indigenous woman from Chilapa, Rita Manuela, who was the sole person not dressed in a fully European style, wearing 'the quaint costume of those of her race' (German y Vázquez 1889, vol. 1, p. 312). In any event, whether because she was indigenous or elderly, the Pope gave her special treatment, leaving the rest of the pilgrims moved 'to see him clasp the Indian woman from Chilapa in his arms and accept with marked complacence the gift she gave him of some Mexican coins with a value of 100 pesos' (German y Vázquez 1889, vol. 1, p. 327).

In Rome, the pilgrims were received by the Bishop of San Luis de Potosí and alumnus of the Pontifical Latin American College, José María Ignacio Montes de Oca. In line with other Mexican and European pilgrimages (Seiler 2007), the report is dotted with descriptions in which feelings are amplified and even presented almost as episodes of ecstasy in which reason is suspended. Thus, for example, the reporter relating the arrival into Rome by rail described how:

> A unanimous exclamation of admiration, joy, and religious exhilaration left the lips of all the Mexicans when they saw approaching the City of the Popes, the Metropolis of Christendom. –Rome! –Rome! cried a hundred voices. –Rome! –Rome! the echo answered back, making hearts beat violently. –Mexico is arriving in Rome! thought I: Mexico is coming to Rome for the first time: Mexico is coming to the centre of the Catholic union. (German y Vázquez 1889, vol. 1, p. 278)

The descriptions became even more intense when narrating the feelings engendered by the presence of the Pope himself and which demonstrate the great importance that devotion to the pope was acquiring in the world. For example, during mass in the Sistine Chapel, the Mexican pilgrims had the opportunity to see Leo XIII for the first time and receive communion from him. In this type of report, the presence of the pontiff was usually an excuse to reflect on the pope's exceptional power as well as his global reach.

These considerations faded away, however, when they approached the Pope to commune, and the reporter changed register to describe how 'my legs grew weak; painfully aware of my insignificance, I understood the greatness of the mercy that I was to receive and, awestruck and bewildered, I came to the feet of the Holy Father, most ardently praying the *Confiteor Deo*' (German y Vázquez 1889, vol. 1, p. 284). In his letters to the newspaper *El Pueblo Católico*, the priest José María Velázquez emphasised the 'anxiety' to 'meet personally the most famous man of our times; the most perfect sage, the Holy Father, before whom even the greatest potentates of the Earth today bend their knee' (Velázquez 1890, p. 95). Velázquez's report is very interesting as it focuses on the pontiff's actual physical appearance, indicating that, while the portraits of him that they had in Mexico are older, 'they are very alike' (96). In this respect, it should be recalled that until Pius XI, the faithful did not have photographs of the pontiff and that it was specifically under him that the image of the pope was portrayed on a multitude of supports in a real *piononomania* (Veca 2018). The proliferation of these images enabled the faithful to make an emotional and symbolic connection with the pope and because of that, affirmations like those of Velázquez reinforced this collective imagination.

After the mass, Leo XIII invited them to a brief reception in the consistory room, where the Mexican pilgrims could enjoy more direct contact with the Pope. A recurring theme of warmth and paternal consolation appears in the descriptions of this encounter, when the reporter evokes the moment in which, kneeling at the feet of Leo XIII, 'feeling his caresses, I felt like a child in the arms of my own father; I felt small in body and spirit; I found myself bewildered beneath the weight of that greatness that has no superior on Earth' (German y Vázquez 1889, vol. 1, p. 291).

The private audience on 14 May provided the climax of the narrative, since the pilgrims could come even closer, receive his blessing, and enjoy a short time with the Pope in which they gave him the gifts, collections, and mementos they had brought him. The Mexican pilgrims prepared themselves beforehand in the Pontifical Latin American College so that they knew how to conduct themselves in the presence of the Pope. Reports are full of the usual ecstatic reactions of the pilgrims in the presence of the pontiff and the acts of kindness they received from him. After a private audience with the Bishops of Chilapa and San Luis de Potosí, the Pope entered with a cortege of bishops and six cardinals, among whom was an old acquaintance of the Latin American Church, the Polish cardinal Ledochowski. The Pope was received with shouts of 'Long live the Holy Father! Long live Leo XIII! Long live the Pope King'! (German y Vázquez 1889, vol. 1, p. 138).

Before the pontiff, the president of the pilgrimage, the Bishop of Chilapa, delivered a speech in which he reflected on the loyalty to the pope of the 'more than nine million Catholics who form the majority of our beloved and Catholic nation, which suffers the tragic exclusion of many of our disgraced compatriots, snatched from the motherly bosom of our holy Church by modern errors' (German y Vázquez 1889, vol. 1, p. 321). After his words, Leo XIII gave them a brief and considered address in which he praised the pilgrims for such a long journey and positioned them through their geographical and social co-ordinates as an American, Mexican, and interclass pilgrimage, 'a most select gathering of American pilgrims who have come here to take part in the festivities of our Sacerdotal Jubilee, representing all the trades and classes of the Mexican Catholic nation' (German y Vázquez 1889, vol. 1, p. 324). The Pope transformed their long journey into the greatest testament to their faith, since only allegiance to the 'Vicar of Christ could move you to traverse the seas and confront the hardships and dangers of such a long voyage' (ibid.). In his speech, he also made mention of the critical situation that the Mexican Church had experienced, its recovery in recent years, and the attention with which he had always watched what was going on in the Mexican Church.

Finally, the Pope underlined the importance of the 'Shrine of Our Lady of Guadalupe, where the most august Virgin, venerated with special reverence by the Mexican people, seems to hold your nation in her gentle protection and lovingly to preserve you in the shelter of her mighty patronage' (German y Vázquez 1889, vol. 1, pp. 324–25). Such

utterances won over an audience that had a strong devotional bond with the advocation of Our Lady of Guadalupe. As the priest Velázquez said, 'it is necessary to be Mexican, and to have known the bitterness of hearing the authenticity of the marvellous image of our national Patron disavowed, to experience the sweet emotion that we felt' (Velázquez 1890, p. 100).

### 2.2.2. The Second National Pilgrimage to Rome (1898)

The second pilgrimage to Rome did not attract such attention and only drew 20 clergy and 17 pilgrims of both sexes. Among the possible reasons are the excessive cost of a pilgrimage that also went to the Holy Land and the fact that it was prompted by no particular celebration, such as the jubilee in the previous pilgrimage. The principal innovation was that it was organised by a lay organisation, the Apostleship of the Cross, along with the Primate of Mexico, Próspero María Alarcón. This Mexican association had been founded in 1895 by the prominent laywoman, Concepción Cabrera de Armida, with the support of the Bishop of Chilapa, Ramón Ibarra y González. In his pastoral in preparation for the pilgrimage, Ibarra himself connected the work of the Apostleship of the Cross, which had been blessed by Leo XIII himself, with the veneration of the pope as an *alter Christus*, who had suffered in their flesh the persecutions of the changing times (Ibarra 1897). In his pastoral preparing for the pilgrimage, the Prelate of Cuernavaca, Fortino Hipólito Vera, also dwelt on the centrality of the pope and Rome by celebrating how 'four decades spent decatholicising Mexico have not destroyed the sacred bond that unites us with the Holy See' (Vera 1897, p. 4).

Of critical importance in the organisation of the pilgrimage was the assistance of the clergy trained in the Pontifical Latin American College in Rome, who thus had broad transnational experience. From Mexico, the pilgrimage was led by the Bishop of Tamaulipas and former student of the Pontifical Latin American College, Filemón Fierro, and the Bishop of Tabasco, Perfecto Amézquita, while Ramón Ibarra had left two months previously to arrange the audience with Leo XIII. The route of the pilgrimage varied slightly as, rather than departing from New York, it did so from Havana in February 1898, just missing the sinking of the USS Maine that triggered the Spanish-American War. Likewise, before arriving in Rome, the ship stopped in Cadiz, Gibraltar, Barcelona, and Montserrat. From Rome they went to the Holy Land and on their return passed through Rome again, stopping along the route in other important Marian sanctuaries such as Loreto, Lourdes, and Zaragoza. Both on the way out and on the way back, the two bishops and some of the pilgrims stayed at the Pontifical Latin American College.

After attending Mass in the Sistine Chapel, Father Trinidad Basurto, parish priest of Calimaya and pilgrimage reporter, described the profound emotional impact caused by the audience with the Pope: 'it was not possible to withstand the feelings and emotions that we experienced, only to express them in some way' (Basurto 1898, vol. 1, p. 139). The pilgrims were accompanied at that point by the Mexicans resident in Rome, the consul, 28 Mexican students at the College, as well as some Argentinians and Bolivians. Abroad, in a country which spoke another language, national differences seemed to fade and a common identity among Latin Americans was reinforced. After the words from the Bishop of Chilapa, Ramón Ibarra, Leo XIII again underscored the importance of this type of demonstration of devotion from such remote regions, expressing 'his gladness at a pilgrimage that came from so far away and that proposes visiting the Holy Land' (*La Civiltà cattolica*, vol. II, 10–23 March 1898).

### 2.2.3. The Third National Pilgrimage to Rome (1900)

The final pilgrimage of the nineteenth century took place on the occasion of the ordinary jubilee of 1900. This celebration became one of the great festivals of Catholicism on a global scale, with 1,300,000 pilgrims in Rome, and it was presented as the triumph of Catholicism in the face of a fateful century which had impeded the celebrations of ordinary jubilees in 1800, 1850, and 1875. The echoes of the news of the opening of the Holy Door

at Christmas in 1899, an act that inaugurated the jubilee, 'spread across the world with admirable speed' and 'was translated into different languages across the whole face of the Earth, awaking the peoples from the indifference in which they lay to proclaim Christ King and absolute ruler over all creation' (Bianchi 1901, pp. 7, 8).

The driving force behind the third pilgrimage was the Catholic printer, Timoteo Macías, who wrote to the Mexican prelates to obtain their support, and once received, approached the Secretary of State, Cardinal Rampolla, to obtain Rome's blessing. With the support of the journalist and playwright, Alberto G. Bianchi, Macías set up offices for the pilgrimage and from there undertook an intense propaganda campaign in the country's main media: *El País*, *La Voz de México*, *El Tiempo*, *El Nacional*, etc.

The pilgrimage was led by the Bishop of Chilapa, Ramón Ibarra, who again acted as mediator between the two spaces of Mexico and Rome. Although they did not find the 500 pilgrims of Macías' ambition, the pilgrimage drew greater numbers than the two previous ones, with 225 pilgrims. In terms of the demographics, it stands out that the great majority were laymen, with a significant presence of women (77 men, 75 women, 11 children), while the number of clergy was less than half, 62 pilgrims in total. As with the previous two, the pilgrims congregated in the capital where they took part in a triduum to the Virgin of Guadalupe in her sanctuary. The train was sent off to cheers and the chords of the Mexican national anthem.

On this occasion, the pilgrims went by train to Veracruz, where the steamboat *Alfonso XIII* awaited them, which had already taken the Argentinian and Uruguayan pilgrims to Rome. Both pilgrimages enjoyed varied hospitality from the Spanish Catholic businessman, the Marquess of Comillas, Claudio López Bru, and his Spanish transatlantic shipping line which connected the Mexican ports with Europe. The Marquess of Comillas had organised and financed a Spanish workers' pilgrimage some years previously, in 1894.

Before going on to Rome, the pilgrimage stopped off at several ports in the north of Spain, going round the entire Iberian Peninsula from Santander to Barcelona. In Cadiz, they met the Argentinian and Uruguayan pilgrimage which was returning from Rome. The warm meeting between the Auxiliary Bishop of Buenos Aires, Gregorio Ignacio Romero, and the Bishop of Chilapa, Ramón Ibarra, symbolised, according to the reporter, a 'brotherly embrace' that united 'on the soil of Mother Spain two of the high dignitaries of the Catholic Church from the two countries which today are at the vanguard of Latin America: the Argentinian and Mexican Republics' (Bianchi 1901, pp. 53, 54).

Upon arrival in Rome, the pilgrims were taken to the Pontifical Latin American College to celebrate a festival in honour of the patron of Mexico, the Virgin of Guadalupe. Visits to this centre increasingly became an almost obligatory stop, first for clergy and later for Latin American pilgrims and visitors. It was necessary for the future of the college that clergy and lay people understood the services it offered. In turn, the centre increasingly became a space to welcome Latin American pilgrims, where they could prepare their visit to the pope and organise all kinds of religious functions.

To gain the jubilee indulgence, the Mexican pilgrims attended the four major basilicas, with the Bishop of Chilapas, Ramón Ibarra, celebrating Mass in St Peter's and the Te Deum in St John Lateran and St Mary Major. During their stay in Rome, the pilgrims visited a place rich with symbolism, the Roman catacombs. Nineteenth century apologetics had used this topos to connect the martyrdom of the early Church under the Roman Empire with other sufferings that Catholics had experienced since then. For the Mexicans there was no doubt: the catacombs were symbolically connected to the situation the Catholic Church in Mexico had recently experienced during the period of the Reform War (1858–1861) and the presidency of Sebastián Lerdo de Tejada (1872–1876). The pilgrims therefore asked for pardon before the martyrs' reliquaries for the sins of the Revolution: 'we recognise, Lord, the sins that Mexico has committed during this century that is about to end, provoking Your righteous anger. We beseech You, Lord, to forgive them all, through the intercessions of Our most gentle Mother, the Virgin of Guadalupe' (Bianchi 1901, p. 123).

The high point of the pilgrimage was the private audience with Leo XIII. The reporter once again emphasised the pope's demonstrations of tenderness and the special treatment that the pontiff had reserved for them. The narration is once again full of portraits of the emotions stirred by his presence, describing how the impression made by the visit would help to pass on devotion to the pope in future: 'the years will pass, and if God grants us life, we will tell our sons and daughters, in our evenings at home, of the blessed interview that the illustrious reigning Pope deigned to offer us, and we will teach them to venerate the Father of all the faithful' (Bianchi 1901, p. 128).

The Mexican pilgrims, with the seminarians from the Pontifical Latin American College, enjoyed a second audience alongside other pilgrimages from other parts of the world. Those present competed in their demonstrations of joy, and the Mexicans sang the Hymn to Our Lady of Guadalupe while the Pope left, borne on his sedia gestatoria, giving his blessing to the audience. The pilgrims again visited the Pontifical Latin American College, taking part in a religious ceremony with students, teachers, and the rector. The seminary was presented to the pilgrims, emphasising especially its contribution to the union of Latin American Catholics:

> The close union of a race that pursues the most noble ideals. Here they look upon each other as true brothers, as members of a single great family, the Mexicans, Central Americans, Venezuelans, Ecuadorians, Colombians, Peruvians, Bolivians, Brazilians, Argentinians, Chileans, and Uruguayans. May all the people of the Latin race one day know the harmony that reigns between the students of this favoured establishment, called to offer inestimable service to the future! (Bianchi 1901, p. 139)

After the Holy Door was closed at the end of the Jubilee and Christmas celebrations, the pilgrims were even able to enjoy a third audience. The Bishop of Chilapa, Ramón Ibarra, spoke and thanked the Pope for his care of 'the distant regions of America', asking his blessing 'on all the faithful of the Mexican nation, especially the indigenous race so devoted to the Church and Our Lady of Guadalupe'. Finally, and as part of the evolution of a Catholic Hispanicism, Ibarra asked for his blessing also for Spain which 'gave us faith and Christian civilisation', so 'that she may continue offering the fruits of faith, holiness, and heroism that covered her in such glory in past centuries' (Bianchi 1901, p. 167). In his response, the Pope praised the virtue of the pilgrims who had come from so far away, and told them that, therefore, 'you have had a special audience with me, which has not been given to others' (Bianchi 1901, p. 172).

## 3. Discussion

The pilgrimages to Rome during the papacy of Leo XIII, and especially the Latin American ones, constitute an area that has been insufficiently explored by historiography. While the number of pilgrims may seem small, it is not so if we compare it with other collective pilgrimages such as the German pilgrimage of 1881 with 500 pilgrims, or with other transnational religious and political gatherings such as the 1215 delegates, who attended the World Missionary Conference in Edinburgh in 1910, or the 334 delegates who attended the International Socialist Labour Congress of Brussels in 1891. In fact, only the workers' pilgrimages organised from France and Spain achieved a truly mass character and they did so because they did not cost as much as a transatlantic pilgrimage and were financially supported by employers and Catholic circles. The numbers of the Mexican pilgrimages take on added value when one takes into account that they are the most important national pilgrimages organised from America in this period, as there were no collective pilgrimages to Rome from the USA, Canada, and other Latin American countries, with the exception of Brazil, Argentina, Uruguay, and Venezuela. Moreover, it should be noted that the impact of these manifestations lies in the fact that they take place alongside other collective and individual pilgrimages that underline the centrality of Rome in global Catholicism. Moreover, in the countries of origin, the impact of such events lies not so much in the number of pilgrims as in the media coverage and publications celebrating

them, and in the awakening of a sense of belonging to a larger entity, the global Catholic Church.

The Mexican pilgrimages contributed to the hierarchisation, centralisation, and globalisation of nineteenth century Catholicism. The reports, pastorals, speeches, and letters analysed reveal the interiorisation and evolution of Ultramontane positions, as well as the successful emotional connection of Mexican Catholics with the pope. In this respect, the Mexican pilgrimages are at the same time a consequence of the process of Romanisation of national churches, and the cause of an even deeper connection of the faithful with Rome. This study corroborates from a Latin American perspective the studies that point to the development of a global devotion to the papacy (Zambarbieri 1990; Horaist 1995; Viaene 2002; Pollard 2005; Rusconi 2010) and, in particular, the emotional and physical connection with the papacy (Seiler 2007). The historiographical contribution of the study of Latin America can therefore be fundamental to understanding the transnational and transatlantic dimension of Ultramontanism, which until then had had an essentially national and European dimension (Viaene 2008; Blaschke 2015), with some exceptions (D'Agostino 2004; Ramón Solans 2020c). This perspective would also make it possible to connect the fertile Latin American national historiographies with their European counterparts and to do so, not as an appendix, but as a fundamental and constitutive part of the development of the Catholic Church in the nineteenth century (Di Stefano and Ramón Solans 2021).

The Mexican pilgrimages, alongside others from countries in Europe and beyond, contributed to endowing the jubilees and other festivals celebrated in Rome with a global character. In summer 1900, *La Civiltà cattolica* celebrated the arrival of pilgrims from such distant places as Argentina, Brazil, Canada, and Venezuela. With the exception of the small Venezuelan pilgrimage led by the Bishop of Calabozo, Felipe Neri Sendra, the rest of the Latin American expeditions included over a hundred people (Serie XVII vol. XI, fasc. 1203, 27 July 1900).

Secondly, pilgrimages reinforced a Catholic reading of Latin America which was fostered on both sides of the Atlantic and that sought to find solutions to the challenges faced by the churches in the different Latin American nations (Ayala Mora 2013; Cárdenas Ayala 2018; Ramón Solans 2020b). This sort of collegiality among the Latin American clergy can be observed in the promotion of the Pontifical Latin American College in Rome during the pilgrimages, as well as the role this centre played in the preparation of audiences with the pope and the hosting of religious ceremonies organised by the pilgrimage. The friendly encounter between clergy from Mexico, Argentina, and Uruguay in Cadiz in the contexts of their respective national pilgrimages for the Jubilee of 1900 likewise illustrates the good will that existed between the Latin American episcopate after the Plenary Council of Latin America.

Finally, the pilgrimages reinforced the image of a Mexican national Catholicism that had not only survived the secularising policies of the 1860s and 1870s, but had also emerged stronger, reformed, and Romanised. The celebration of those national pilgrimages reveals the dynamism of a highly internationalised clergy, which had contributed to the reform of the Mexican dioceses, as well as the presence of a dynamic and active lay movement, which collaborate in the organisation and propaganda of these events. The experience of the pilgrimage, as well as the words and favours from the Holy See, strengthened their Catholic national pride. Finally, during the pilgrimages, the Mexican national symbol par excellence, the Virgin of Guadalupe, remained linked with Rome; the Virgin appeared as the pilgrims' protector on such a long journey—an endeavour recognised upon their arrival in Rome, to the pride of the participants, by Leo XIII himself.

## 4. Materials and Methods

The historical study of individual pilgrimages has traditionally been based on the study of pilgrims' registration and accommodation records, their individual narratives, official documents issued by the authorities, etc. The collective nature of the pilgrimages studied means that the sources are different, since they tended to generate more official

documentation in the form of reports and memorabilia. The character of the public demonstration of the pilgrims' faith and of the collectivity that they claimed to represent means that many of those sources were printed and used for propaganda.

This article has drawn on the official reports of the three national pilgrimages, as well as on pastorals, speeches, published correspondence, and echoes in the Roman press. In terms of authorship, laymen played a central role by writing the reports of the first and third national pilgrimage. Although unsigned, the first was the work of the law graduate and collaborator with Catholic conservative newspapers such as *La Voz de México*, Diego German y Vázquez who registered his authorship with the Mexican Secretary of State (Recopilación 1890, p. 859). The narrative of the third national pilgrimage was the work of the Catholic journalist and dramatist Alberto G. Biancchi. The rest of the documents used were produced by clergy of varying ranks, from the pastoral signed by the prelates Ramón Ibarra and Fortino Hipólito Vera to the letters by the cathedral canon, León José María Velázquez, and the account of the second Mexican pilgrimage by the presbyter, José Trinidad Basurto.

Analysis of these documents has taken into account their official character and the wish to magnify the nature of these events. The focus is not, however, so much on the reliability of these narratives as on the sense they give of these events and their desire to emphasise the figure of the pope and the religiosity of the pilgrims and the Mexican nation. In this respect, the descriptions fit with those studied by other researchers (Zambarbieri 1990; Horaist 1995; Seiler 2007; Rusconi 2010) since the Mexican narratives were inspired by the news that arrived of other pilgrimages and the documents and announcements made by the Vatican, as well as the many testimonies of devotion to the pope. All this reinforces the idea of the globalisation and homogenisation of Catholic discourses in the nineteenth century.

## 5. Conclusions

At the start of the turbulent nineteenth century, the distance between a believer and the Holy See was enormous, both physically and symbolically. It was not only the enormous cost of time and money that caused a displacement of these features. The pope was a distant figure, whose role within the Catholic Church was unknown by the majority of the faithful who could not even imagine how he looked. His global authority was hampered by a network of royal privileges, and local traditions mediated devotional practices. At the end of the century, all this had changed, with new modes of transport and communication as well as political and ecclesiastical changes bringing the pope closer to the faithful. The Roman question and the Culture wars favoured the appearance and development of mass global demonstrations in support of the papacy, among which stand out the national pilgrimages to Rome. The presence of pilgrimages that had come from such distant places as Mexico contributed to reinforcing the global authority of the papacy. Closeness to the pope was not only physical but also emotional; the pope became the visible head of the Catholic Church, the leader of global Catholic opinion.

The Mexican national pilgrimages were the result of the reform of the national church towards Rome, the implementation of new devotional cultures, and the internationalisation of Mexican clergy. These changes were not the result of an imposition or mere passive import but were the product of the specific political and religious situation in Mexico, as well as the initiative of some local religious actors who sought inspiration beyond their borders to confront the problems they were facing. To make a pilgrimage from Guadalupe to Rome thus represented an assertion of the nation's Catholicity, while also demonstrating the strength and global nature of papal power.

**Funding:** This research was funded by the Spanish Ministry of Science, Innovation and Universities under the Ramón y Cajal research program, grant number RYC2019-026405-I. Research support from the Spanish Ministry of Education and Science through research projects HAR2015-65991-P and PID2019-108299GB-C21 is gratefully acknowledged.

**Conflicts of Interest:** The author declares no conflict of interest.

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
