# Peer review of "‘A Most Select Gathering’. Mexican National Pilgrimages to Rome during the Papacy of Leo XIII"

_religions, doi:10.3390/rel12070475_

Round 1
Reviewer 1 Report
The article builds on the state of the art and addresses a clearly identified gap in the literature. A first part provides the necessary context of the situation of the Catholic Church in 19th century Mexico. The study of the three Mexican pilgrimages is informative and draws on a variety of sources. Overall, the article succeeds at showing how the reports of individual pilgrims as well as official documents that originated in the context of the organized large-scale pilgrimages can be contextualized in a discussion of Church politics and Mexican identity, and the wider issues of globalisation and homogenisation of Catholicism during the 19th century. The article is a good fit for the proposed special issue. I recommend it for publication.
I only have some very minor suggestions:
The section named “Results” (238) provides the historical context for the understanding of the three pilgrimages, and this could be reflected in the heading.
In line 238 I assume it should be Colosseum instead of Coliseum.
There are some inconsistencies with the quotation marks (467, 472ff). While the language is generally good, the article might benefit further from a critical reading by a native speaker.
Author Response
Dear reviewer,
Thank you for your feedbacks and suggestions, which I have already corrected.
Reviewer 2 Report
The article gives fruitful insights and puts the Mexican pilgrims in a broader context of Latin-American pilgrimages. On the other hand it relies on very subjective reports of euphemistic clerical and lay participants who use stereotypes and contemporary semantic cliches in order to express themselves. The author tends to exaggerate at some important points in the article:
The main argument is a bit exaggerated:
“The Mexican pilgrimages contributed to the hierarchisation, centralisation, and globalisation of nineteenth century Catholicism.”
In fact, only a handful of people came, and at the third pilgrimage at least 500 pilgrims were planned to go to Rome, in the end they only managed to attract 225 people. Why was there not more financial support? Why was the pilgrimage to Rome so unattractive?
The resume is another exaggeration, drawing a huge conclusion from poor evidence:
“The celebration of those national pilgrimages reveals the dynamism of a highly internationalised clergy [...] as well as the presence of a dynamic and active lay movement.” Where is the dynamic and active lay movement, if at the first pilgrimage only 150 people went there (not from the masses, as the author emphasises), in the second only 17 laymen (plus 20 clerics) and on the last one only 225 instead of the expected 500? The conclusion doesn’t harmonise with the evidence of the article.
Another exaggeration is that the “transnational and transatlantic dimension of Ultramontanism” in former studies“ had had an essentially national and European dimension (Viaene 2008; Blaschke 2015) with some exceptions (D’Agostino 2004; Ramón Solans 2020c)” – until this important and “fundamental” study of this author.
True is that there are few studies concerning the transatlantic dimensions of ultramontanism. But there are some which indeed go beyond the national and European dimension and are more than mere exceptions:
Arx, Jeffrey P. (Ed.) Varieties of Ultramontanism. Washington, D.C. 1998.
Blaschke, Olaf / Francisco Javier Ramón Solans (Ed.), Weltreligion im Umbruch. Transnationale Perspektiven auf das Christentum in der Globalisierung, Frankfurt 2019.
De Groot, Kees. Brazilian Catholicism and the Ultramontane Reform. West Lafayette, 2003.
Oliveira, Gustavo de Souza. Aspectos do ultramontanismo oitocentista: Antonio Ferreira Viçoso e a Congregação da Missão. Tese de Doutorado. Departamento de História. Universidade Estadual de Campinas. Campinas, 2015.
Prien, Hans-Jürgen, Christianity in Latin America. Revised and Expanded Edition, Leiden 2013.
Santirocchi, Ítalo Domingos. ma questão de revisão de conceitos: Romanização – Ultramontanismo – Reforma. In: Temporalidades (Belo Horizonte), v. 2, n. 2, 2010.
Santirocchi, Ítalo Domingos. Questão de consciência: os ultramontanos no Brasil e o regalismo do Segundo Reinado (1840-1889). Belo Horizonte 2015 (mentioned, but not in this context).
Small remarks:
The article claims to be also based on sermons (p. 13) but doesn’t give a single example.
Why are “Argentina, Brazil, Canada, Croatia, and Venezuela”, four in America and one close to Italy, “distant places”? What does Croatia do in this list?
Author Response
Dear reviewer,
Thank you for your suggestions and comments. I have introduced the small changes you suggested regarding Croatia and the sermons, there is a mistake as I was referring to pastorals.
Regarding the question of exaggerations, I have introduced this paragraph in which I compare with other religious manifestations of the time:
"While the number of pilgrims may seem small, it is not so if we compare it with other collective pilgrimages such as the German pilgrimage of 1881 with 500 pilgrims or with other transnational religious and political gatherings such as the 1,215 delegates who attended the World Missionary Conference in Edinburgh in 1910 or the 334 delegates who attended the International Socialist Labour Congress of Brussels in 1891. In fact, only the workers' pilgrimages organised from France and Spain achieved a truly mass character and they did so because they did not cost as much as a transatlantic pilgrimage and were financially supported by employers and circles. The numbers of the Mexican pilgrimages take on added value when one takes into account that they are the most important national pilgrimages organised from America in this period, as there were no collective pilgrimages to Rome from the USA, Canada and other Latin American countries with the exception of Brazil, Argentina, Uruguay and Venezuela. Moreover, it should be noted that the impact of these manifestations lies in the fact that they take place alongside other collective and individual pilgrimages that underline the centrality of Rome in global Catholicism. Moreover, in the countries of origin, the impact of such events lies not so much in the number of pilgrims as in the media coverage and publications celebrating them and in the awakening of a sense of belonging to a larger entity, the global Catholic Church."
With regard to the laity, they are not only involved in the pilgrimage, but also in the organisation and promotion.
With regard to the question of ultramontanism, the sentence alludes to the lack of studies from a transnational perspective. Of course, I am familiar with the studies you mentionned and I could add more on Peru, Argentina or Chile by Sol Serrano, Roberto Di Stefano, Ignazio Martinez or Jeffrey L. Klaiber. In order to avoid confusion and hurt sensitivities I have added
"This perspective would also make it possible to connect the fertile Latin American national historiographies with their European counterparts and to do so not as an appendix, but as a fundamental and constitutive part of the development of the Catholic Church in the nineteenth century."
Best regards